# Copper (II) Species with Improved Anti-Melanoma and Antibacterial Activity by Inclusion in β-Cyclodextrin

**DOI:** 10.3390/ijms24032688

**Published:** 2023-01-31

**Authors:** Alina Tirsoaga, Victor Cojocaru, Mihaela Badea, Irinel Adriana Badea, Arpad Mihai Rostas, Roberta Stoica, Mihaela Bacalum, Mariana Carmen Chifiriuc, Rodica Olar

**Affiliations:** 1Department of Analytical and Physical Chemistry, Faculty of Chemistry, University of Bucharest, 4-12 Regina Elisabeta Av., District 3, 030018 Bucharest, Romania; 2Department of Inorganic and Organic Chemistry, Biochemistry and Catalysis, Faculty of Chemistry, University of Bucharest, 90-92 Panduri Str., District 5, 050663 Bucharest, Romania; 3National Institute for Research and Development of Isotopic and Molecular Technologies, Department of Physics of Nanostructured Systems, 67-103 Donat Str., 400293 Cluj-Napoca, Romania; 4Horia Hulubei National Institute for Physics and Nuclear Engineering, Department of Life and Environmental Physics, 30 Reactorului Str., 077125 Magurele-Ilfov, Romania; 5Department of Microbiology, Faculty of Biology, University of Bucharest, 1-3 Aleea Portocalelor Str., District 5, 060101 Bucharest, Romania; 6Romanian Academy of Scientists, 54 Spl. Independenței Str., District 5, 050085 Bucharest, Romania; 7Biological Sciences Division, The Romanian Academy, 25 Calea Victoriei, Sector 1, District 1, 010071 Bucharest, Romania

**Keywords:** antibacterial activity, Cu(II) complex, β-cyclodextrin, inclusion system, melanoma, molecular docking

## Abstract

To improve their biological activity, complexes [Cu(bipy)(dmtp)_2_(OH_2_)](ClO_4_)_2_·dmtp (**1**) and [Cu(phen)(dmtp)_2_(OH_2_)](ClO_4_)_2_·dmtp (**2**) (bipy 2,2′-bipyridine, phen: 1,10-phenantroline, and dmtp: 5,7-dimethyl-1,2,4-triazolo [1,5-a]pyrimidine) were included in β-cyclodextrins (β-CD). During the inclusion, the co-crystalized dmtp molecule was lost, and UV-Vis spectra together with the docking studies indicated the synthesis of new materials with 1:1 and 1:2 molar ratios between complexes and β-CD. The association between Cu(II) compounds and β-CD has been proven by the identification of the components’ patterns in the IR spectra and powder XRD diffractograms, while solid-state UV-Vis and EPR spectra analysis highlighted a slight modification of the square-pyramidal stereochemistry around Cu(II) in comparison with precursors. The inclusion species are stable in solution and exhibit the ability to scavenge or trap ROS species (O_2_·^−^ and HO·) as indicated by the EPR experiments. Moreover, the two inclusion species exhibit anti-proliferative activity against murine melanoma B16 cells, which has been more significant for (**2**)@β-CD in comparison with (**2**). This behavior is associated with a cell cycle arrest in the G0/G1 phase. Compared with precursors, (**1a**)@β-CD and (**2a**)@β-CD exhibit 17 and 26 times more intense activity against planktonic *Escherichia coli*, respectively, while (**2a**)@β-CD is 3 times more active against the *Staphylococcus aureus* strain.

## 1. Introduction

Cancer is the second leading cause of death worldwide. Among skin malignities, melanoma is characterized by rapid growth and spreading, for which current therapies are challenged by resistance and increased risk of infection [1]. Even if current treatments still use both organic and inorganic cytostatics, they are expensive and exhibit limited effectiveness, especially for recurrent and metastatic diseases.

For instance, melanoma treatment includes either dacarbazine as a single agent or carboplatin combined with paclitaxel as combinatory organic cytostatic therapy [2]. However, both therapies lead to severe side effects, including acquired resistance coupled with increased infection risk. Consequently, challenges and opportunities remain in developing new chemotherapeutics for melanoma treatment or in developing appropriate carriers to increase the effectiveness of the current treatments.

By contrast, inorganic cytostatics such as cisplatin and its analogs, well-known worldwide as species used for treating some primary tumors, have limited use due to the resistance development and a restricted activity domain [3,4]. Therefore, one of the current strategies to develop new anticancer therapeutics is focused on other metallic ions that could exhibit different mechanisms of action and/or an extended spectrum of activity compared with platinum-based cytostatics. In recent years, good potential from this point of view was observed for Cu(II) complexes that can inhibit angiogenesis, tumor growth, and metastasis by a combined and complex mechanism that involves DNA intercalation or coordination and reactive oxygen species (ROS) generation. Such compounds exhibit nuclease-like activity leading to DNA cleavage either by phosphate ester hydrolysis and/or by deoxyribose and nucleobase moieties oxidation [5]. The DNA denaturation occurs mainly through hydroxyl radicals generated either in a Fenton or in Haber–Weiss-type reactions [6] produced by hydrogen peroxide reduction under the influence of a Cu(II) complex that exhibits SOD-like activity. In addition, it is worth mentioning that if a ligand substituent can establish π–π stacking or other non-covalent interactions with DNA strands, then the ability of the complex to link and break the DNA chain is increased [5,6].

Although these compounds exhibit good anticancer activity and some have been proven active in platinum-insensitive tumor cells, only two Cu(II) species have reached the preclinical studies stage [7]. Nevertheless, despite promising preclinical results, attempts to develop these compounds as drugs have unraveled some inconveniences. For instance, their low water solubility leads to the decrease of accumulation inside tumor cells and tissues. In order to improve this feature, species sequestration in some organic matrixes, such as cyclodextrins (CDs), has been proposed as a strategy [7,8]. Recently, CDs were approved by Pharmacopoeia as excipients used in several pharmaceutical products as drug delivery systems. This ability is based on their safe toxicological profile and capability to form water-soluble complexes. In aqueous media, CDs can form either an inclusion complex with a lipophilic molecule or a part of it or a non-inclusion complex [9,10].

Concerning the CDs, the common natural CDs have six (α-CD), seven (β-CD), or eight (γ-CD) glucopyranose units, exhibiting different water solubility. Due to the chain arrangement of the glucopyranose units, CDs are shaped similar to cones, with the secondary hydroxyl groups extending from the wider edge and the primary ones from the narrow edge [10,11,12,13]. Moreover, it has been shown that CDs increase the water solubility of poorly soluble organic and inorganic species and, thus, their bioavailability and stability. This strategy has already been successfully used for the development of about 40 organic drug delivery systems, including anticancer agents [14]. Most anticancer species associated with CDs have shown improved solubility and stability, increased bioavailability and dissolution, reduced toxicity, and enhanced delivery [12,15,16,17,18,19].

Even if one can speak about many forms containing various CDs, especially β-CD and its derivatives, as of yet there is no therapeutic formulation on the market containing an inorganic anti-cancer drug prepared with CDs [20]. 

On the other hand, available research concerning the copper (II) based complexes’ inclusion in CDs is very scarce. However, these evidence the ability of some species to interact with such a matrix. Thus, the [Cuddtc_2_] (Hddtc = diethyldithiocarbamate acid) is a potent anticancer agent against a wide range of tumors (glioblastoma, solid tumors involving the liver, prostate cancer). Still, it exhibits low solubility in water [21], which has been improved by complex inclusion in hydroxypropyl-β-CD or sulfobutyl ether-β-CD. Both were proven as promising systems against sensitive and chemoresistant breast cancer cell lines [22]. The preclinical toxicity studies in rats and rabbits revealed the high safety profile of [Cu(crc)(OH)(OH_2_)] (crc = curcumin) included in β-CD after topical vaginal application [23].

Moreover, a functionalized fabric with eco-friendly and sustainable performance was prepared by Cu(II) coordination with a material, resulting in the multi-crosslinking of citric acid to poly(trimethylene terephthalate) and β-CD. This species induced a growth inhibition of 90.75% in *Escherichia coli* and 85.36% in the *Staphylococcus aureus *strains [24]. These data suggest the opportunity to exploit the utility of CDs for developing delivery systems for Cu(II) biologically active agents in the future.

Correspondingly, the complexes [Cu(bipy)(dmtp)_2_(OH_2_)](ClO_4_)_2_·dmtp (**1**) and [Cu(phen)(dmtp)_2_(OH_2_)](ClO_4_)_2_·dmtp (**2**) (bipy 2,2′-bipyridine, phen: 1,10-phenantroline, and dmtp: 5,7-dimethyl-1,2,4-triazolo[1,5-a]pyrimidine) (Figure 1) were reported by our research group as species with good anti-melanoma and antimicrobial activity [25].

In an attempt to develop a delivery system assuring an improved therapeutic potential, these complexes were included in β-CD in molar ratios of 1:1 and 1:2, as was deduced from Jobs curves and docking studies. The new materials were characterized in solid state and in solution concerning spectral features, stability, and ability to interact with ROS. Additionally, their anti-melanoma and antibacterial activity were assessed and compared with precursors that lose the co-crystalized molecule of dmtp during interaction with β-CD. To sum up, these materials are superior to their precursors regarding their enhanced water solubility and improved bioavailability of active species—aspects reflected in the increased activity against both melanoma cells and the tested pathogenic bacteria.

## 2. Results and Discussion

Current studies demonstrate that the anticancer strategy based on copper complexes–CD inclusion systems may represent a suitable and selective strategy for developing efficient and safe new drugs. For the success of such formulations, the complex active framework and CD derivative set are of crucial importance, since they can modulate stability, the lipophilic/hydrophilic balance of the resulting system, solubility in biological fluids, as well as the ability to permeate the lipid membrane.

The presence of hydrophobic parts in the complexes [Cu(bipy)(dmtp)_2_(OH_2_)](ClO_4_)_2_·dmtp (**1**) and [Cu(phen)(dmtp)_2_(OH_2_)](ClO_4_)_2_·dmtp (**2**) was studied by testing their ability to form inclusion species with β-CD. Their formulations with β-CD were prepared by simply mixing the components in a water–ethanol mixture at room temperature. During the synthesis, it was observed that the co-crystalized molecule of dmtp was lost, and as a result the included complexes are
[Cu(bipy)(dmtp)_2_(OH_2_)](ClO_4_)_2_@β-CD abbreviated as (**1a**)@β-CD
[Cu(phen)(dmtp)_2_(OH_2_)](ClO_4_)_2_@β-CD abbreviated as (**2a**)@β-CD

### 2.1. Formulation and Characterization of Inclusion Complexes

#### 2.1.1. Stoichiometry of Inclusion Complexes

Job’s method of continuous variation adapted for inclusion equilibrium involving chemical species and β-CD was used in this part of the study [26]. As β-CD didn’t show absorption spectra in the UV-Vis region, preliminary information about the spectral behavior of the complexes and dmtp needed to be obtained. For this purpose, the spectra of the solution-containing complex, dmtp, and the complex with β-CD in a ratio of 1:5 versus dmtp were recorded, but neither (**1**) nor (**2**) showed a significant difference in the shape of the spectra. However, an important difference in the absorbance value for (**1**) and (**2**) at 312 and 272 nm, respectively, showed the need to develop Job’s method to find out the stoichiometry of the host-guest equilibrium. Job plots obtained for systems (**1**)–β-CD and (**2**)–β-CD are presented in Figure 2. As one can observe, the curve in Figure 2a indicates for the molar fraction of complex (R) a value of 0.5 corresponding to a 1:1 stoichiometry between (**1a**) and β-CD, while that presented in Figure 2b shows 0.33 for R, which corresponds to a 1:2 molar ratio between (**2a**) and β-CD.

#### 2.1.2. Molecular Docking Studies

A molecular modeling approach was performed to explore the putative binding of β-CD with chemical species (**1a**) and, respectively, (**2a**), as a theoretical assessment of the experimental results obtained by both Job’s method and phase solubility studies that indicate the formation of supramolecular associations of 1:1 for (**1a**)@β-CD and 1:2 for (**2a**)@β-CD, respectively.

The geometry of both (**1a**) and (**2a**) species has been used from experimental single-crystal investigations previously reported by our group [25] without further optimization. The β-CD (published as the CCDC #762697 structure) geometry was optimized at PM6-D3H+ level [27,28] to correct both dispersion and intra-molecular hydrogen bond interactions.

The molecular docking calculations were performed with the AutoDock-Vina engine [29,30] using box sizes up to 20·20·20 Å and a spacer of 0.375 Å (exhaustiveness parameter of 32). The putative binding of (**1a**) to β-CD (negative binding energy of about −3.10 kcal/mol) consists of an inclusion complex with the dmtp fragment (Figure 3a) with 1:1 stoichiometry. Attempts to dock to the dipy fragment side exclude this possibility, resulting in an unrealistically high (positive) binding energy. The binding of (**2a**) to β-CD presents a different behavior: putative binding poses indicate as viable possibilities both inclusion complexes: with the dmtp fragment (similar to the previous case, −3.45 kcal/mol) but also with the more prominent phen fragment. However, slightly less energy stabilizes the former pose (−2.96 kcal/mol). Both negative binding energy values obtained for these (**2a**) to β-CD interaction modes suggest the possibility of a sequential formation of a 1:2 (**2a**): β-CD supramolecular association (Figure 3b).

#### 2.1.3. Characterization of Inclusion Complexes in Solid State

##### Powder XRD Studies

Powder X-ray diffraction provides useful information that identifies a certain type of compound through a characteristic diffraction line pattern.

The characteristic peaks for β-CD appears at 2θ values of 9.1, 16.1, 18.9, and 22.8° (marked with * in Figure 4a). The diffractogram of (**1a**)@β-CD (Figure 4b) shows that this species exhibits a lower crystallinity. Still, the peak’s characteristic for β-CD (marked with *) can be observed together with supplementary ones at 2θ of 7.4, 19.9, and 23.3°, which can be assigned to the core of complex (**1**) (marked with •). The powder XRD of complex (**1**) simulated from single crystal data [25] is presented in Figure 4c. From Figure 4, it can be seen that some signals are broadened as a result of peaks from both precursors and are overlapped.

The β-CD characteristic values can also be observed for the (**2a**)@β-CD species (marked with * in Figure 5b) completed with new ones at 2θ of 7.7, 8.0, 8.8, 13.1, 15.1, 18.8, and 22.5° that are characteristic for the core of complex (**2**) (marked with •) (Figure 5c).

Thus, the presence of β-CD and complex (**1**) and (**2**) signals in diffractograms demonstrates the formation of inclusion species through physical interactions.

##### FTIR Spectra

The most important IR bands in β-CD and the inclusion complexes’ IR spectra can be seen in Figure 6 and are summarized in Appendix A.

A large number of intense bands can be observed in the 3200–3550 cm^−1^ range in the IR spectrum of β-CD (marked with * in Figure 6a), assigned to the stretching vibrations of the hydroxyl groups. The same groups are responsible for the band at 1645 cm^−1^ assigned to bending vibration modes [10,11].

The spectra of inclusion complexes (Figure 6b,c) exhibit the characteristic pattern of both β-CD and complexes (**1**) and (**2**). Hence, the band associated with the ν(OH) vibration of the hydroxyl groups are broader, with the maximums at 3386 and 3405 cm^−1^, respectively. At the same time, the band assigned to δ(OH) is shifted from 1645 to 1629 cm^−1^ due to the interaction with the Cu(II) complexes. The supplementary band around 1550 cm^−1^ can be assigned to the ν(C=N) of dmtp, while those at 1089/1078 and 624 cm^−1^ indicate the free perchlorate presence in the inclusion species [31]. These bands (marked with • in inclusion complexes spectra) are slightly shifted compared to the precursor (**1**) and (**2**) spectra [25], except the band assigned to the stretching vibration of perchlorate ν_3_ is moved from 1090 to 1078 cm^−1^ for (**2a**)@β-CD as a result of the interactions with the CD hydroxyl groups.

##### Solid-State UV-Vis Spectra

A comparison between the UV-Vis spectra of the inclusion complexes’ spectra and that of the precursor Cu(II)-based complexes also indicate a slight modification of stereochemistry upon interaction with β-CD. In the spectra of precursor complexes (**1**) and (**2**), a wide band with maxima at 600 and 590 nm can be noticed. The band is shifted to 665 nm for both inclusion complexes (Appendix A). Its feature, with a shoulder at higher wavelength numbers, is characteristic of the same square planar stereochemistry [32] identified by single crystal X-ray data for precursor complexes [25].

##### EPR Spectroscopy on Powder Samples

To evaluate the paramagnetic properties of complexes (**1a**)@β-CD and (**2a**)@β-CD, EPR measurements both in X- and Q-band were performed. The results are presented in Figure 7a,b, revealing an EPR spectrum characteristic for Cu(II). The EPR powder spectrum of (**1a**)@β-CD shows an axial g-tensor with g_ǁ_ = 2.25 and g_⊥_ = 2.06 and a solved hyperfine structure with A_ǁ_ = 495 MHz. In contrast, the powder EPR spectrum of (**2a**)@β-CD shows a slight isotropic g-value with g = 2.064, and no hyperfine structure is observable, indicating that the molecular and crystallographic axes are misaligned [33]. When the ground state of the Cu(II) center is the d_x_^2^_−y_^2^ orbital, elongated octahedral, square pyramidal, or square planar geometries are expected. In this case, the EPR spectra are axial (Figure 7a,b), with g_ǁ_ (g_z_) and g_⊥_ (g_x_ = g_y_) where g_ǁ_ > g_⊥_ > g_e_ = 2.0023 [34], which are in good agreement with the observed g values and the reported structure of the two Cu(II)-based complexes [25].

#### 2.1.4. Characterization of Inclusion Complexes in Solution

##### Phase Solubility Study

The phase solubility diagrams of complexes (**1**) and (**2**) are shown in Figure 8. Compound (**1**) follows a linear profile described in the literature as an A_L_ type, according to Higuchi and Connors [35]. This behavior corresponds to a molar ratio of 1:1 between (**1a**) and β-CD, similar to that obtained using Job’s method. Further on, the value of Kc was calculated using the phase solubility diagram and was found to be 250 M^–1^. This value confirms that the equilibrium between (**1**) and β-CD shifts to the product, indicating a high tendency of (**1**) to enter the cavity of β-CD together with good stability of the (**1a**)@β-CD complex. On the other hand, the phase stability curve for (**2**) (Figure 8b) shows a shape that was fitted using a quadratic polynomial model. Higuchi described this model as being A_P_, meaning a positive deviation from the linearity, corresponding to a 1:2 molar ratio between (**2**) and β-CD, in very good agreement with results obtained by Job’s method. This type of stoichiometry literature indicates a possibility of evaluating the stability constant starting from the equation of a quadratic model [36]. Even if this is not a perfect model and the value calculated is affected by an error ~30%, reliable information about the stability of the (**2a**)@β -CD complex could be obtained. The model ensures the calculation of K_1:1_ and K_1:2_. The obtained values were 20.44 and 6 M^–1^. In addition, as seen in Figure 8, both (**1**) and (**2**) were solubilized in water in the presence of β-CD.

##### Solution EPR Spectra

The inclusion compounds were dissolved in DMSO, and the solution EPR spectra are presented in Figure 9. At the same time, the stability of the compounds in DMSO, after two weeks, was evaluated using EPR spectroscopy. The spectra of the two compounds do not change (Figure 9) and show remarkable stability, which is important for biological tests.

A temperature sweep experiment was performed to better understand the mobility and gain more information about the g-tensor of the compounds. Thus, the temperature varied from 260 to 316 K in 4 K steps, and the EPR spectrum was measured for each step (Figure 10a,b). Since a 100 mM (in DMSO solution) concentration of the compounds was used, the strong exchange interaction observed in the powder spectra of (**2a**)@β-CD (Figure 7) disappears, and both spectra at 260 K show a well-resolved hyperfine interaction A_ǁ_. The EPR parameters (g-tensor, hyperfine coupling tensor, and line widths) presented in Table 1 were determined after simulations (Figure 10c,d) of the spectra obtained at 260 K using an EasySpin’s [37] “pepper” simulation routine for solid-state continuous wave EPR spectra.

Both compounds show similar line broadenings and g- and A-values. The g-tensor is rhombic and is not an axial one, where g_x_ = g_y_ ≠ g_z_, which indicates that the ground state of the Cu(II) is a linear combination between the d_x_^2^_−y_^2^ and d_z_^2^ orbitals. For complexes of this type, the parameter R can be indicative of the predominance of the d_z_^2^ or d_x_^2^_−y_^2^ orbital in the ground state, where R is defined as (g_y_ − g_z_)/(g_x_ − g_y_) with g_x_ > g_y_ > g_z_. If R > 1, the greater contribution to the ground state arises from the d_z_^2^ orbital; if R < 1, the greater contribution to the ground state arises from the d_x_^2^_−y_^2^ orbital. In our case, R is 0.02 for the compound (**1a**)@β-CD and 0.06 for (**2a**)@β-CD, indicating that the contribution from the d_x_^2^_−y_^2^ orbital is predominant. Thus, elongated octahedral, square pyramidal, or square planar geometries are expected, which is in good agreement with the single-crystal XRD data where a square pyramidal stereochemistry was reported for free Cu(II) complexes [25].

The simulation of the EPR spectra recorded at 308 K reveals the mobility of the compounds in x, y, and z directions, which are given by the correlation times obtained with the “chili” simulation routine for slow-motion continuous-wave EPR spectra (Table 1). The results reveal that compound (**1a**)@β-CD has higher mobility in the y direction. In contrast, compound (**2a**)@β-CD is more mobile in the x direction, with correlation times in the ns range.

The ability of the inclusion compounds to scavenge or trap ROS was tested using EPR spectroscopy, for which KO_2_ (100 mM) and H_2_O_2_ (3%) were used as O_2_·^−^ and OH·-radical sources, respectively. A 1 mM DMSO solution was mixed with the radical donors for each experiment. Figure 11a,b shows the EPR spectra of (**1a**)@β-CD and (**2a**)@β-CD before and after ROS exposure. Both compounds (**1a**)@β-CD and (**2a**)@β-CD show trapping abilities for O_2^−^_ radicals, as indicated by the changed EPR signal intensity of the g_z_ component, which indicates that the inclusion in β-CD makes the sixth position around Cu(II) more accessible for superoxide coordination.

At the same time, no scavenging or trapping potential for the OH· radicals were observed for (**1a**)@β-CD since the EPR signal presents no changes after the H_2_O_2_ addition. Compound (**2a**)@β-CD shows scavenging abilities for OH· radicals, as indicated by the increased EPR signal intensity aspect not observed at precursor complex (**2**) [25].

### 2.2. Anti-Melanoma and Antibacterial Activity

#### 2.2.1. Anti-Melanoma Activity

The viability of L929 fibroblasts and B16 melanoma cells was evaluated for the new compounds after 24 and 48 h of treatment (Figure 12). The compound (**1a**)@β-CD and β-CD did not affect the viability of the normal fibroblast cells after 24 or 48 h of treatment with concentrations up to 25 µg/mL (Figure 12a,c). However, compound (**2a**)@β-CD started to affect L929 cell viability significantly from a concentration of 0.39 µg/mL, decreasing cell viability from 93.88 to 10.93% at the highest concentration tested after 24 h of treatment (Figure 12a) and from a concentration of 0.78 µg/mL, for a viability of 96.20 to 8.13% at the highest concentration tested after 48 h of treatment (Figure 12c). The IC_50_ values calculated from the viability curves were 6.88 µg/mL (2.27 μM) at 24 h and 4.97 µg/mL (1.64 μM) at 48 h. In previously reported studies, the complex (**2**) used as a precursor for (**2a**)@β-CD synthesis had an IC_50_ value of 13.61 µg/mL (15.04 µM) [25], indicating that the new compound is more toxic for normal cells compared with its precursor.

However, the effects of the two compounds on the melanoma cells show slightly higher toxicity than on normal cells. Compound (**1a**)@β-CD starts to affect cell viability from a concentration of 0.78 µg/mL at both investigated treatment times (Figure 12b,d). At the highest-tested concentration of 25 µg/mL (8.25 µM), the viability of the tumoral cells decreased to 83.87% after 24 h and to 62% after 48 h of treatment. The IC_50_ values were 6.50 µM after 24 h and 4.42 µM after 48 h of treatment, while in a previous study [25] involving precursor (**1**), no IC_50_ could be calculated. The results indicated that the new compound, (**1a**)@β-CD, is less toxic for B16 cells. For compound (**2a**)@β-CD, similar to L929 cells, the viability of the melanoma cells decreased proportionally with the concentration (Figure 12d). When the treatment was applied for 24 h, the cell viability decreases significantly at concentrations higher than 3.12 µg/mL (Figure 12b). When the treatment was applied for 48 h, the cell’s viability decreased from 91.73% at 0.39 µg/mL to 6.91% at 25 µg/mL (Figure 12d). The IC_50_ values were 5.98 µg/mL (1.97 µM) after 24 h and 2.92 µg/mL (0.96 µM) after 48 h of treatment. Compared with precursor (**2**) [25], which had IC_50_ values against B16 cells of 6.88 µg/mL (7.60 µM) after 24 h and 4.15 µg/mL (4.58 µM) after 48 h of treatment, the new compound is more toxic to melanoma cells.

Based on the cell viability results, we can state that compound (**2a**)@β-CD shows slightly higher toxicity against melanoma cells compared to normal cells, as one can observe from the calculated therapeutic index (TI) (Table 2). The TI was 1.15 after 24 h and 1.70 after 48 h of treatment. Compared with the previously reported TI for precursor (**2**) [25], the values were 1.98 at 24 h and 1.08 at 48 h, which indicates that compound (**2a**)@β-CD has a higher specificity for melanoma cells compared to precursor (**2**) after 48 h of treatment.

Previous results have indicated that copper(II) complexes’ anti-proliferative effect is correlated with oxidative DNA lesions [38,39,40,41,42]. An efficient technique to evaluate the DNA damage caused by the newly synthesized complexes is to monitor their influence on cell cycle progression. The progression of different cell cycle phases is highly regulated by a series of checkpoints, ensuring that only the cells with undamaged DNA can enter cell division [43,44]. One checkpoint regulates cell transition from the G1 phase to the S phase, where DNA synthesis starts, and a second one regulates cells transition from the G2 phase to the M phase, when the two daughter cells separate by mitosis [43,44].

The influence of the cellular cycle was investigated for two concentrations—1.56 µg/mL, a concentration below the IC_50_ value obtained for compound (**2a**)@β-CD, and 6.25 µg/mL, a concentration close to the IC_50_ value for the same compound.

The percentages of cells found in each phase for both L929 and B16 cells are presented overlapped in Figure 13. For L929 untreated cells, the percentage of cells found in the G0/G1 phase is 52.14 ± 1.44%, 22.46 ± 1.43% for cells in the S phase, and 22.14 ± 1.41% in the G2/M phase (Figure 13). When treated with compound (**1a**)@β-CD at a lower concentration (1.56 μg/mL), the cell cycle distribution of the L929 cells is not affected (Figure 13). However, when treated with a higher concentration of 6.25 μg/mL, the number of cells found in the S phase decreased significantly to 17.9 ± 0.82% (*p* < 0.05), while the ones in the G2/M phase increased to 27.0 ± 1.23% (*p* < 0.05), with no changes in the G0/G1 phase.

When fibroblast cells were treated with compound (**2a**)@β-CD at a lower concentration, the cells found in the G0/G1 phase decreased to 46.42 ± 0.66% (*p* < 0.005), and the ones in the G2/M phase increased to 29.0 ± 3.39% (*p* < 0.0005). Surprisingly, at the higher concentration of 6.25 μg/mL, the number of cells found in the G0/G1 phase increased to 69.72 ± 2.07% (*p* < 0.0001), and that of the ones found in the S phase decreased to 10.13 ± 1.64% (*p* < 0.0001), with no change in the G2/M phase compared with the control condition.

For untreated melanoma cells, the percentage of cells found in the G0/G1 phase is 59.66 ± 1.94%, in the S phase 16.76 ± 3.75%, and in the G2/M phase 21.28 ± 21.28% (Figure 13). When compound (**1a**)@β-CD was applied, no significant changes were induced, irrespective of the tested concentration, although a small increase of the cell number found in the G2/M phase was observed (23.81 ± 4.86 %). When compound (**2a**)@β-CD was applied, the number of cells found in the G0/G1 phase increased to 64.66 ± 3.15% after treatment with 1.56 μg/mL and to 68.60 ± 4.29% (*p* < 0.0001) after treatment with a concentration of 6.25 μg/mL.

For both cell lines, β-CD did not significantly alter the number of cells in either cell cycle phase.

Based on the results reported above, when treated with compound (**2a**)@β-CD at the highest concentration, DNA damage in tumoral melanoma cells is detected, and the G1 checkpoint is activated, leading to an extension of the G0/G1 phase, which is indicative of an anti-proliferative effect and apoptosis. Similar results (G0/G1 phase arrest and apoptosis induction), both for B16 cells as well as for other cancer cell types, were previously reported to be induced by different copper(II) complexes [45].

#### 2.2.2. Antibacterial Activity

Previous studies show that metabolic changes associated with malignancy as well as an anticancer treatment could favor the occurrence of dysbiosis and opportunistic infections. Hence, if the antitumor species could also exhibit an antimicrobial activity directed towards the main opportunistic bacterial pathogens, this could represent an advantage and be associated with a lower risk of side effects and complications.

The antibacterial activity of the inclusion complexes was evaluated on the Gram-negative *Escherichia coli* ATCC 25922 and Gram-positive *Staphylococcus aureus* ATCC 25923 reference bacterial strains. The antibacterial activity of the β-CD, precursor complexes, and inclusion complexes was evaluated against planktonic and biofilm-embedded cells to obtain the minimum inhibitory (MIC) and minimum biofilm eradication (MBEC) concentrations, respectively.

The β-CD exhibited no activity on the tested strains. Compared to the bare compounds ((**1**) and (**2**)), the antibacterial activity of their inclusion species was significantly improved for both species in the case of *E. coli* and compound (**2**)@β-CD in the case of *S. aureus* (Table 3).

Although complexes (**1**) and (**2**) exhibit antibiofilm activity, their inclusion species did not show such behavior, probably because bulk inclusion complexes cannot penetrate the biofilm matrix [46].

Such antimicrobial potential was also observed for a zinc and titanium oxide nanoparticle based on multishell hollow spheres—material that can be also used for drug delivery [47]. Moreover, the same team reported a method based on fluorescence measurement that allows the rapid detection and counting of bacteria by using water-soluble carbon quantum dots as a fluorescence marker [48].

## 3. Materials and Methods

### 3.1. Materials and Physical Measurements

Chemicals were purchased from Sigma-Aldrich (St. Louis, MO, USA) (copper(II) perchlorate hexahydrate (≥99.99% trace metals basis), 1,10-phenantroline (phen, 99%) 2,2′-bipyridine (bpy, 99%)), Sigma-Aldrich (Darmstadt, Germany) (2,3-pentanedione (97%), and 3-amino-4H-1,2,4-triazole (96%)), Merck (Darmstadt, Germany) (dibenzo-18-crown-6-ether, potassium superoxide) and MP Biomedicals (Illkich, France) (β-CD) in reagent grade. All these were used as received without further purification. Complexes (**1**) and (**2**) were prepared as reported [25].

Fourier Transform Infrared spectroscopy (FTIR) spectra were recorded in KBr pellets with a Tensor 37 spectrometer (Bruker, Billerica, MA, USA) in the 400–4000 cm^−1^ range. UV-Vis spectra in solid state were recorded using a Jasco V 670 spectrophotometer (Jasco, Easton, MD, USA) with Spectralon as a standard in the 200–1500 nm range, while UV-Vis spectra in aqueous solution were recorded using a Jasco V 530 spectrometer controlled by Spectra Manager software. Quartz cells of 1 cm were used for all measurements. Powder X-ray diffraction patterns were obtained with a Bruker D8 Advance X-ray diffractometer (Bruker, Karlsruhe, Germany) (Cu anode and Ni filter, λ = 1.54184 Å) in Bragg-Brentano configuration. X and Q-band EPR measurements were carried out with a continuous wave dual-band E500 ELEXSYS EPR spectrometer (Bruker, Karlsruhe, Germany). The room temperature measurements in X-band were carried out at a microwave frequency of 9.879 GHz, whereas the measurements in Q-band were at 34.16 GHz. The low-temperature X-Band measurements were carried out with an ER4131 liquid nitrogen temperature controller (Bruker, Karlsruhe, Germany). The microwave frequency, in this case, was 9.45 GHz. The free radical scavenging ability of the complexes was tested through EPR spectroscopy using KO_2_ and H_2_O_2_ as sources for O_2_^−^ and OH^−^. KO_2_ was dissolved in DMSO by complexation with dibenzo-18-crown-6-ether.

### 3.2. Synthesis of Inclusion Complexes

The inclusion complex (**1a**)@β-CD was prepared as follows: to a solution containing complex (**1**) (0.240 g, 0.1 mmol) in 10 mL ethanol, a solution containing β-CD (0.568 g, 0.5 mmol) in 10 mL water was added. The reaction mixture was magnetically stirred at 50 °C for 48 h until a yellow precipitate (dmtp) was formed. The green solution was evaporated at room temperature, 30 mL of methanol was added, and the excess β-CD was removed by filtration. The obtained solution was concentrated, and the formed solid product was filtered off, washed several times with cold ethanol, and dried in an air atmosphere.

The inclusion complex (**2a**)@β-CD was prepared as follows: to a solution containing complex (**2**) (0.226 g, 0.1 mmol) in 10 mL ethanol, a solution containing β-CD (0.568 g, 0.5 mmol) in 10 mL water was added. The reaction mixture was magnetically stirred at 50 °C for 48 h until a yellow precipitate (dmtp) was formed. The green solution was evaporated at room temperature, 30 mL of methanol was added, and the excess β-CD was removed by filtration. The obtained solution was concentrated, and the formed solid product was filtered off, washed several times with cold ethanol, and dried in an air atmosphere.

### 3.3. Study of Host–Guest Equilibrium in Aqueous Solution

#### 3.3.1. Stoichiometry of Inclusion Complexes

Stock solutions of 1 mM were prepared by weighing appropriate quantities of each studied compound, followed by quantitative transfer with distilled water in 50 mL volumetric flasks. First, spectra of a solution containing complex (0.1 mM) and dmtp (0.1 mM) versus dmtp (0.1 mM) and complex (0.1 mM) with β-CD (0.1 mM) in a ratio of 1:5 versus dmtp (0.1 mM) were recorded.

Job’s method was developed according to the literature data [26]. A series of solutions containing a molar fraction ranging from 0 to 1 of each complex was prepared, and UV-Vis spectra were recorded. The difference in absorbance of the complex with and without β-CD noted by ΔA multiplied by the molar fraction of complex (R = [complex]/([complex] + [β-CD]) was plotted versus R. The stoichiometry of the host–guest complex with β-CD was found as the maximum deviation of R.

#### 3.3.2. Phase-Solubility Studies

The method settled for phase-solubility studies was adapted from a previous paper [49]. First, calibration curves for (**1**) and (**2**) were obtained. For this purpose, successive dilutions from the stock solution were performed, spectra were recorded and absorbance reeded 312 nm (**1**) and 272 nm (**2**). The plot of absorbance versus concentration gives two straight lines for both (**1**) and (**2**) complexes: y = 0.5111x − 0.0323 (R^2^ = 0.9996) and y = 18.792x − 0.0323 (R^2^ = 0.9996), respectively. Second, water and β-CD 20 mM were added to six samples containing an excess of complex (50 mg) that had the same volume but increasing β-CD concentration. In all experiments, the final volume was 5 mL. The samples were shaken for 1 h, filtered using 0.25 µm PTFE, and the solution was diluted to 1:25 mL before the spectra were recorded. The concentration of the complex in each solution was calculated using a calibration curve described above. In the end, the phase solubility diagrams were obtained by plotting the concentration of the complex versus the concentration of β-CD. The stability constants were calculated from the phase-solubility diagram as follows: (i) Kc = slope/[So × (1 − slope)], where So is the complex’s solubility in the absence of β-CD, for a 1:1 mole ratio, and (ii) S_Total_ = S_0_ + K_1:1_[β-CD] + K_1:1_K_1:2_[β-CD]^2^, where K_1:1_ and K_1:2_ are constants for 1:1 and 1:2 host–guest complexes and S_Total_ is the total solubility of complex, for a 1:2 stoichiometry.

### 3.4. Computational Strategy

The geometry of β-cyclodextrin was obtained from the CCDC database (#762697, downloaded from https://www.ccdc.cam.ac.uk/; accessed on 22 July 2022). Theoretical calculations were performed using the following software suites: (a) GAMESS 2021 R2 Patch 2 [28] source code compiled using AMD-optimized C-lang v3.2.0 with AOCL extensions v3.1.0 on x86_64 Alma Linux 8.7 and run on a Ryzen 9 gen 4 CPU, (b) AutoDock-Vina v1.2.3-52 source [29,30] compiled statically with GCC 8.5.0 using system default boost libs v1.66 on the same machine, and (c) Python Molecule Viewer v1.5.7p1 as implemented in MGL Tools v.1.5.7 tarball installer [50].

### 3.5. In Vitro Cytotoxicity Assay

#### 3.5.1. Cell Culture Conditions

Mouse fibroblast cell line L929 (CCL-1, ATCC, USA) and mouse melanoma cell B16 (ATCC CRL-6475, USA) were grown in DMEM (Dulbecco’s Modified Eagle Medium) supplemented with 2 mM L-Glutamine, 10% fetal calf serum (FCS), 100 units/mL of penicillin, and 100 µg/mL of streptomycin. They were kept at 37 °C in a humidified incubator under an atmosphere containing 5% CO_2_. All cell-cultivation media and reagents were purchased from Biochrom AG (Berlin, Germany) and Sigma-Aldrich (Darmstadt, Germany).

#### 3.5.2. Cellular Viability Assay

To test cellular viability, we used the MTT assay (3-(4,5-dimethylthiazol-2-yl)-2,5-diphenyltetrazolium bromide) colorimetric assay as previously described [51]. Briefly, L929 and B16 cells were seeded in 96-well plates at a density of 5000 cells/well and grown in the culture medium for 24 h. The next day, the cells were treated with the investigated compounds, added in concentrations of 0.39, 0.78, 1.56, 3.12, 6.25, 12.5, and 25 μg/mL for 24 and 48 h, respectively. The cells cultivated without the tested compounds served as negative controls. After the incubation period, the culture media was removed, and the MTT solution was added at a final concentration of 0.5 mg/mL. This assay is based on transforming the tetrazolium salt into formazan crystals in the metabolically active cells. After 4 h of incubation at 37 °C, the formazan crystals were dissolved using DMSO. The absorbance of the samples was recorded at λ = 570 nm using a plate reader (Mithras 940, Berthold, Bad Wildbad, Germany). The data were corrected for background absorbance, and the viability was calculated using the following formula:% viable cells = [(A_570_ of treated cells)/(A_570_ of untreated cells)] × 100 (%)

Half-maximal inhibitory concentration (IC_50_) was determined by fitting the data with a sigmoidal logistic function using Origin 8.1 (Microcal Inc., Northampton, MA, USA) software.

#### 3.5.3. Cell Cycle Analysis

A flow cytometry analysis was used to check the possible influence of the tested compounds on the cell cycle of both cellular lines L929 and B16. Before treatment, 25,000 cells per well were seeded in 24-well plates and treated with 2 concentrations of the compounds (1.56 and 6.25 µg/mL) for 24 h. After treatment, cells were detached, washed with cold PBS, and then resuspended in cold PBS mixed with 100% ethanol and kept at −20 °C for at least 24 h for fixation. The following day, the cells were stained using a 0.1% Triton-X in PBS containing 0.2 mg/mL RNAase and 20 µg/mL propidium iodide. The samples were incubated for 30 min at 37 °C in the dark. After incubation with the staining solution, the cells were washed with PBS and resuspended in 100 µL PBS. The cell cycle measurement was made using CytoFLEX (Beckman Coulter, Brea, CA, USA) by recording 30,000 events/sample. All experiments were made in triplicate, each having at least 2 wells per condition. The data was evaluated using the CytoFLEX software.

### 3.6. Antibacterial Activity Assay

The antibacterial activity assays were carried out on the Gram-negative *E. coli* ATCC 25922 and the Gram-positive *S. aureus* ATCC 25923 reference strains. The antimicrobial activity of the β-CD inclusion complexes versus the β-CD and Cu(II) complexes against planktonic bacterial cells was assessed using the microdilution assay, allowing MIC determination [25]. Anti-biofilm activity was evaluated using the crystal-violet microtiter to determine the MBEC [25]. All experiments were performed in triplicate. In both assays, serial binary concentrations ranging from 8.81 to 0.013 mM were used.

### 3.7. Statistical Analysis

The experiment was performed at least 3 times with at least 2 replicas per condition/experiment. The data are presented as means ± standard deviations if not stated otherwise. The statistical analysis was carried out using the GraphPad Prism 9 software package (San Diego, CA, USA). Analysis of variance (ANOVA) with Dunnett’s multiple comparisons post-test versus control condition was used to calculate statistical significance. A value of *p* < 0.05 was chosen to indicate that the difference was statistically significant.

## 4. Conclusions

Two Cu(II) complexes with mixed heterocycle N-donor ligands were successfully included in β-CD. Theoretical calculation results suggest the energetically possible formation for two supramolecular associations that differ in stoichiometry (1:1 for (**1a**):β-CD and 1:2 for (**2a**):β-CD) but share a common aspect—the inclusion of one dmtp fragment, in line with both Job’s method and phase-solubility experimental data. At the investigated concentrations, compound (**1a**)@β-CD did not show any effects against the L929 fibroblast and B16 melanoma cells. However, compound (**2a**)@β-CD showed an increased potential as an antitumor agent compared with its precursor (**2**) previously reported. Compound (**2a**)@β-CD’s efficiency against melanoma cells was at least twice as high compared to normal fibroblast cells, and the IC_50_ concentration was 0.96 μM after 48 h of treatment. Furthermore, compound (**2a**)@β-CD can induce DNA damage, which leads the cells to enter G0/G1 cell cycle arrest followed by possible apoptosis. The antibacterial potential was improved against planktonic strains but was quenched in the case of microbial biofilms. Both species exhibit the ability to scavenge ROS species, consequently proving their potential to be developed as anti-inflammatory drugs.

## Figures and Tables

**Figure 1 ijms-24-02688-f001:**
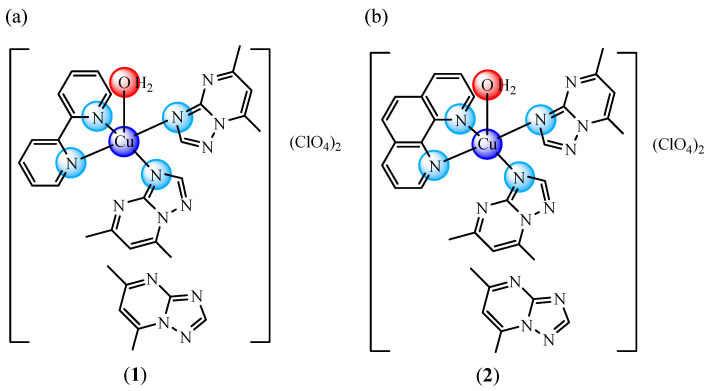
The molecular structure of complexes [Cu(bipy)(dmtp)_2_(OH_2_)](ClO_4_)_2_·dmtp (**1**) (**a**) and [Cu(phen)(dmtp)_2_(OH_2_)](ClO_4_)_2_·dmtp (**2**) (**b**).

**Figure 2 ijms-24-02688-f002:**
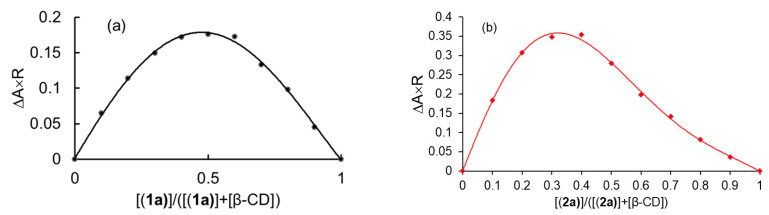
Job’s plots for inclusion complexes (**1a**) (0.1 mM) and β-CD (0.1 mM) (**a**) at 312 nm and (**2a**) (0.1 mM) and β-CD (0.1 mM) (**b**) at 272 nm.

**Figure 3 ijms-24-02688-f003:**
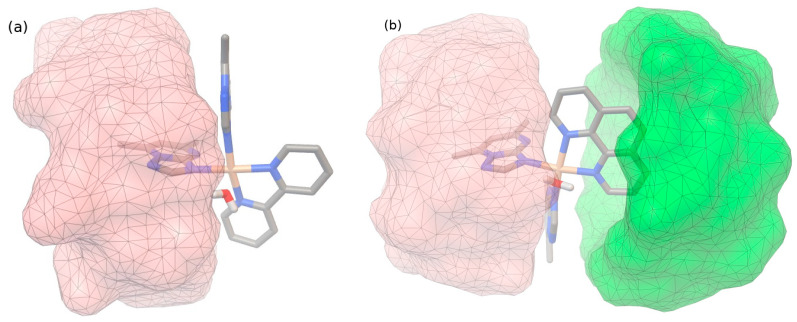
Putative docking result of (**1a**)@β-CD (**a**), and (**2a**)@β-CD (**b**) supramolecular association as simulated with AutoDock-Vina algorithm.

**Figure 4 ijms-24-02688-f004:**
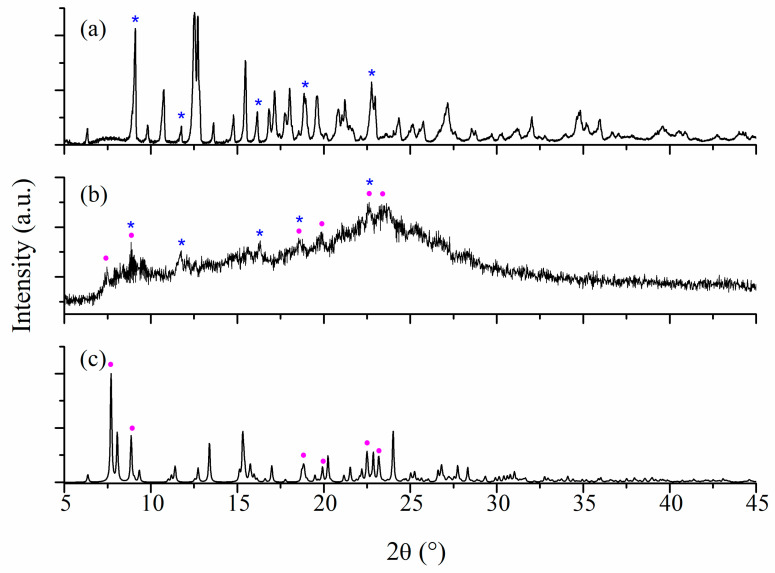
Powder X-ray diffractograms for β-CD (peaks marked with *) (**a**), (**1a**)@β-CD (**b**), and simulated from single crystal X-ray analysis for (**1**) (peaks marked with •) (**c**).

**Figure 5 ijms-24-02688-f005:**
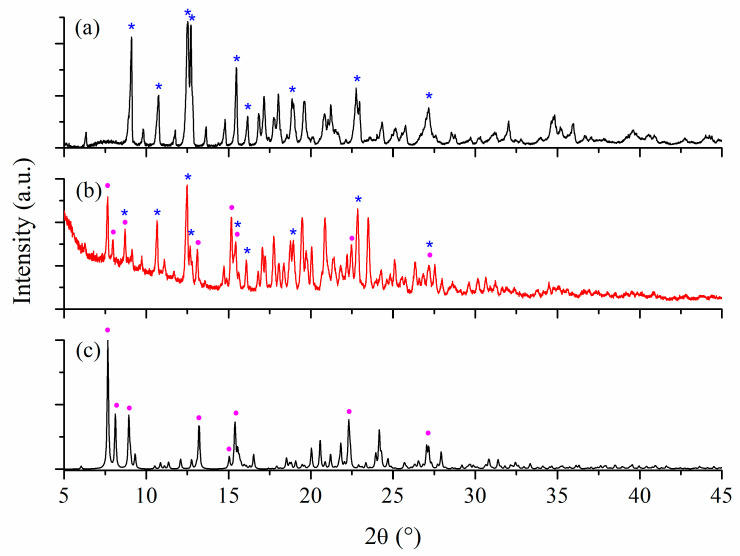
Powder X-ray diffractograms for β-CD (peaks marked with *) (**a**), (**2a**)@β-CD (**b**), and simulated from single crystal X-ray analysis for (**2**) (peaks marked with •) (**c**).

**Figure 6 ijms-24-02688-f006:**
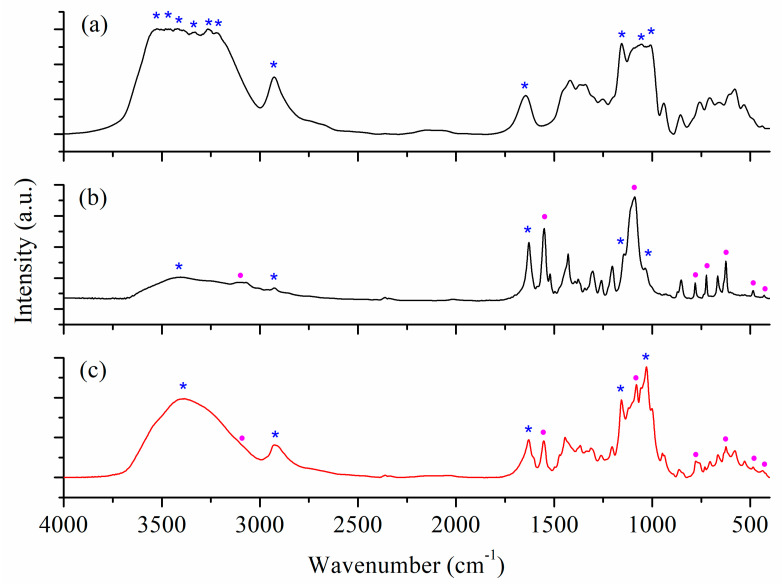
IR spectra of β-CD (bands marked with *) (**a**) and inclusion complexes (**1a**)@β-CD (**b**) and (**2a**)@β-CD (**c**) registered in KBr discs. The bands corresponding complexes (**1**) and (**2**) are marked with •.

**Figure 7 ijms-24-02688-f007:**
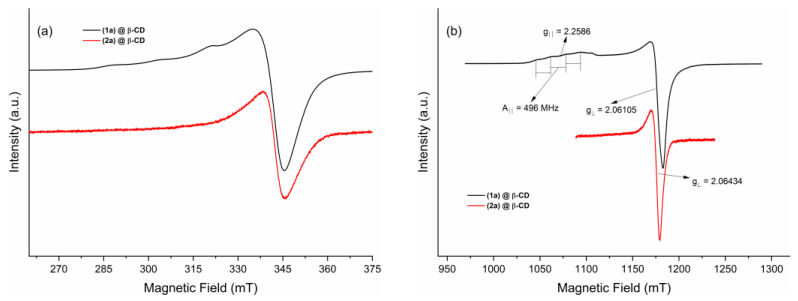
X-band (9.879 GHz frequency) (**a**) and Q-band (34.16 GHz frequency) (**b**) powder EPR spectra of inclusion complexes (**1a**)@β-CD and (**2a**)@β-CD.

**Figure 8 ijms-24-02688-f008:**
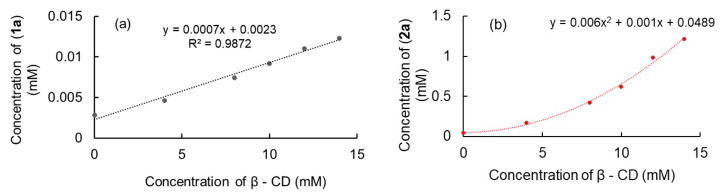
Phase solubility diagrams of (**1**) and β-CD system (**a**) and (**2**) and β-CD system (**b**).

**Figure 9 ijms-24-02688-f009:**
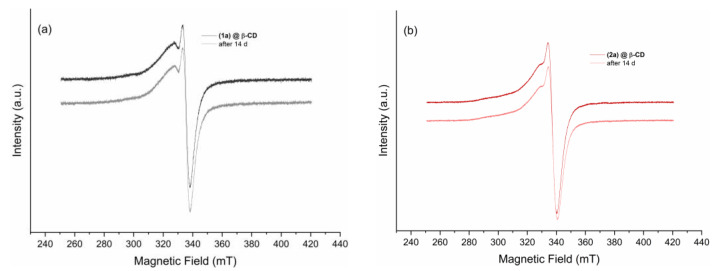
EPR spectra of (**1a**)@β-CD (**a**) and (**2a**)@β-CD (**b**) dissolved in DMSO measured at 9.88 GHz with a 100 mM concentration. Each spectrum was also remeasured after 14 days to evaluate the stability of the two complexes in the solution.

**Figure 10 ijms-24-02688-f010:**
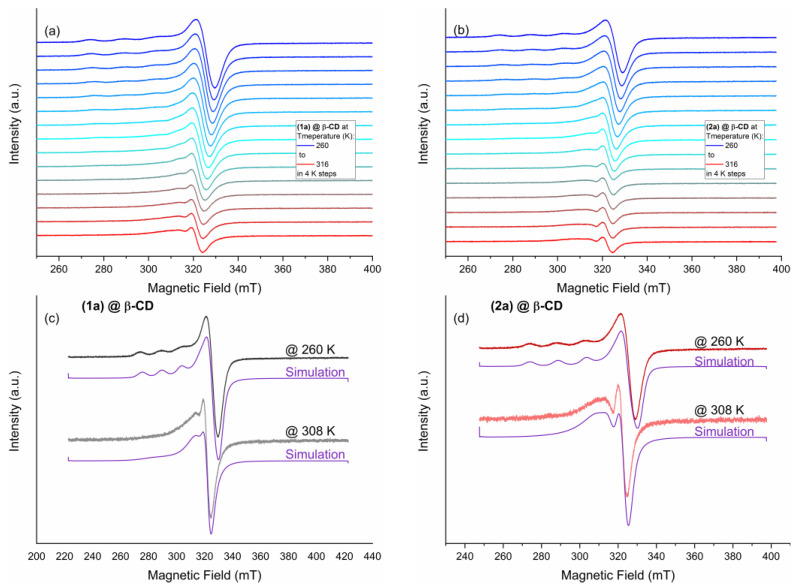
EPR spectra of (**1a**)@β-CD (**a**) and (**2a**) @ β-CD (**b**) dissolved in DMSO with a 100 mM concentration measured from 260 to 316 K with a 4 K step. Simulated EPR spectra together with the measured data recorded at 260 and 308 K for (**1a**)@β-CD (**c**) and (**2a**) @ β-CD (**d**), respectively. All measurements were carried out at a microwave frequency of 9.45 GHz.

**Figure 11 ijms-24-02688-f011:**
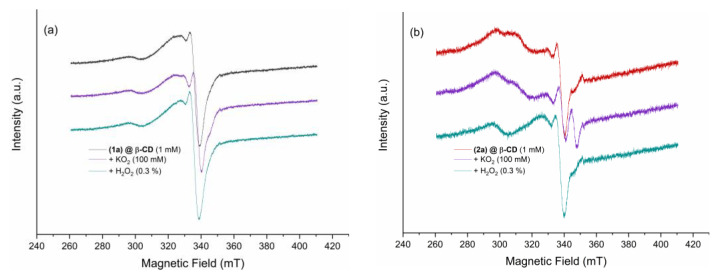
EPR spectra of (**1a**)@β-CD (**a**) and (**2a**)@β-CD (**b**) dissolved in DMSO measured at a microwave frequency of 9.88 GHz with a 1 mM concentration also in the presence of ROS donors KO_2_ (O_2_·^−^) and H_2_O_2_ (OH·).

**Figure 12 ijms-24-02688-f012:**
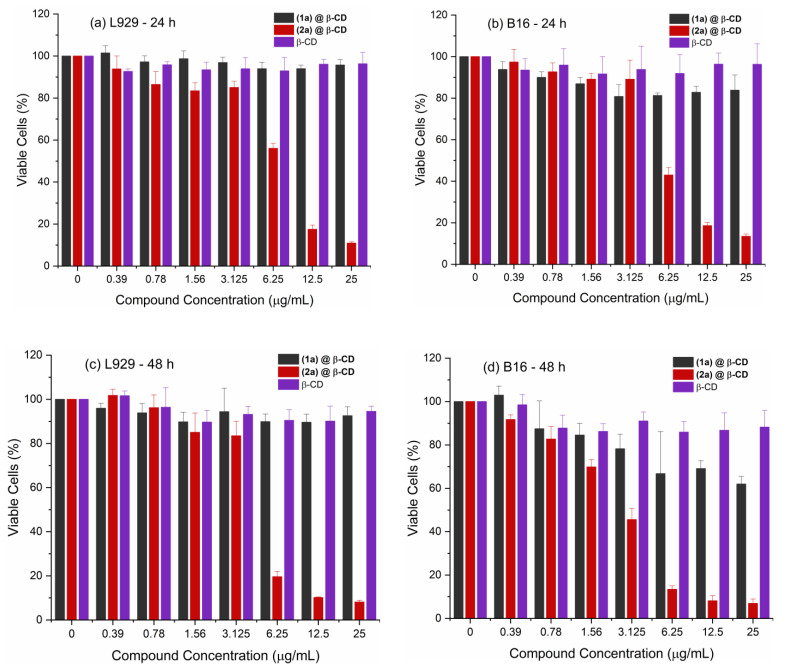
Cell viability of L929 cells treated for 24 h (**a**) and 48 h (**c**), and B16 cells treated for 24 h (**b**) and 48 h (**d**) with increasing concentrations of the three compounds (*p* values are based on ANOVA analysis with Dunnett’s multiple comparisons post-test versus control condition).

**Figure 13 ijms-24-02688-f013:**
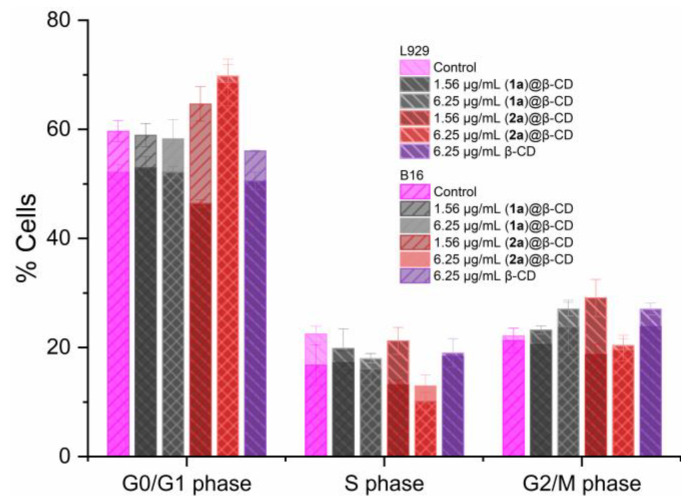
Cell cycle distribution of L929 and B16 cells treated with (**1a**)@β-CD, (**2a**)@β-CD and β-CD for 24 h at 1.56 and/or 6.25 µg/mL concentrations (*p* values are based on ANOVA analysis with Dunnett’s multiple comparisons post-test versus control condition).

**Table 1 ijms-24-02688-t001:** Simulation EPR parameters for (**1a**)@β-CD and (**2a**)@β-CD.

Compound	Temperature (K)	Simulated EPR Parameters			
		g-Tensor	A-Tensor (MHz)	Line Width (mT)	Correlation Times (ns)
		[g_x_ g_y_ g_z_]	[A_x_ A_y_ A_z_]	[Gaussian Lorenzian]	
(**1a**)@β-CD	260	[2.279 2.076 2.072]	[436 5.4 0.7]	[4 2.5]	---
	308			[3 2.1]	[0.02 25.11 0.18]
(**2a**)@β-CD	260	[2.282 2.081 2.069]	[460 5.4 0.7]	[4 1.5]	---
	308			[2.5 1.5]	[9.54 0.19 0.14]

**Table 2 ijms-24-02688-t002:** IC_50_ values and calculated TI for the compounds tested.

Compounds	IC_50_/µg/mL (µM)				TI	
	L929		B16			
	24 h	48 h	24 h	48 h	24 h	48 h
(**1a**)@β-CD	-	-	-	-	-	-
(**2a**)@β-CD	6.88 (2.27)	4.97 (1.64)	5.98 (1.97)	2.92 (0.96)	1.15	1.70
β-CD	-	-	-	-	-	-

**Table 3 ijms-24-02688-t003:** The MIC values (mM) for β-CD, precursors, and inclusion species.

Bacterial Strain	β-CD	(1)	(1a)@β-CD	(2)	(2a)@β-CD
*E. coli* 134202	8.81	0.71	0.04	0.34	0.013
*S. aureus* 25923	8.81	0.18	0.20	0.04	0.013
Ref.	this paper	[25]	this paper	[25]	this paper

## Data Availability

Not applicable.

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
