# Peer review of "Copper (II) Species with Improved Anti-Melanoma and Antibacterial Activity by Inclusion in β-Cyclodextrin"

_ijms, 2023, doi:10.3390/ijms24032688_

Round 1
Reviewer 1 Report
The manuscript presented for consideration is an interesting comprehensive study of copper compounds included in cyclodextrins as antitumor and antibacterial drugs.
The authors have done a great job and certainly the material will be of interest to the readers of our journal.
There are several suggestions on how to improve the manuscript.
1. It is absolutely necessary to give the structural formulas of all the compounds under consideration as Figure 1.
2. In the caption to all figures, all the conditions for the experiment should be indicated.
3. I recommend enriching Figure 3 by including those spectra that are now presented in additional materials. Discussing IR spectra without showing them in the main text is not very convenient.
Reviewer 2 Report
The abstract and introduction should be rewritten. The present form is very confusing and does not link the two sentences properly. Sometimes the sentences seem to start suddenly.
Another hand, In throughout the introduction not mentioned about melanoma, the authors mentioned overall cancer.
The introduction should mention why this material is better than prior works.
Original XRD and FTIR Spectra should provide in n main manuscript with clear marks, changes and etc.
"The data are presented in Figure 8. Compound (1a)@β-CD and 340 β-CD did not affect the viability of normal fibroblast cells after 24 or 48 h of treatment with 341 concentrations up to 25 µg/mL (Figure 8A and C).However, compound (2a)@β-CD starts 342 to affect the cell viability from the concentration of 0.39 µg/mL at both treatment times 343 investigated (Figure 8A and C). " But in figure 8 a-d, all represent near about same results towards viability for (2a)@β-CD. Need clarification.
Need to include some appropriate and recent references like, doi:10.1166/jbn.2011.1344; DOI: 10.1109/TNB.2022.3186941 and etc
Round 2
Reviewer 2 Report
1. The abstract and introduction should be rewritten. The present form is very confusing and does not link the two sentences properly. Sometimes the sentences seem to start suddenly.
Both abstract and introduction were rewritten. The same problem still persists and needs more careful to change.
2. Another hand, In throughout the introduction not mentioned about melanoma, the authors mentioned overall cancer.
The aspects concerning melanoma were provided. The introduction should mention why this material is better than prior works.This aspect was covered in Introduction. English language needs to be checked carefully (native english checker would be preferred).
3."The introduction should mention why this material is better than prior works" .This aspect was covered in Introduction. Authors did not provided the changes like line no, page no etc.Yet what I have examined does not fit the context of this manuscript.
4. Original XRD and FTIR Spectra should provide in main manuscript with clear marks, changes and etc. The powder XRD diffractograms and IR spectra (with characteristic signals for both precursors marked) were provided in main text and Table 1 was moved at Supplementary. As result the Figures and Tables were renumbered. The colour for system (2a)@β-CD was changed in red in all Figures and in Figure 13 the data for cell cycle distribution of L929 and B16 cells were superposed. Change marking was not provided. This is important for easy understanding of the reader. Therefore authors should mark the change peaks with different colors or etc.
5. "The data are presented in Figure 8. Compound (1a)@β-CD and 340 β-CD did not affect the viability of normal fibroblast cells after 24 or 48 h of treatment with concentrations up to 25 µg/mL (Figure 8A and C).However, compound (2a)@β-CD starts to affect the cell viability from the concentration of 0.39 µg/mL at both treatment times investigated (Figure 8A and C). " But in figure 8 a-d, all represent near about same results towards viability for (2a)@β-CD. Need clarification.
These aspects were simplified and clear presented. Authors did not provided any clarifications in modified manuscript. They have not even provided written instructions where the figure number has been changed.
6. Need to include some appropriate and recent references like, doi:10.1166/jbn.2011.1344; DOI: 10.1109/TNB.2022.3186941 and etc The recommended references were not provided because we consider that are not connected with the main aspect presented in paper, complexes included in -CD matrix. Authors should provide appropriate justification for how these references are not appropriate.
Round 3
Reviewer 2 Report
Good